# Hot Deformation Behaviour of Mn–Cr–Mo Low-Alloy Steel in Various Phase Regions

**Ivo Schindler [1],\*** , **Petr Opěla [1]** , **Petr Kawulok [1]** , **Jaroslav Sojka [1]** , **Kateřina Konečná [1]** , **Stanislav Rusz [1]** , **Rostislav Kawulok [1]** , **Michal Sauer [1]** and **Petra Turoňová [2]**

[1] Faculty of Materials Science and Technology, VŠB—Technical University of Ostrava, 17. Listopadu 2172/15, 70800 Ostrava, Czech Republic; petr.opela@vsb.cz (P.O.); petr.kawulok@vsb.cz (P.K.); jaroslav.sojka@vsb.cz (J.S.); katerina.konecna@vsb.cz (K.K.); stanislav.rusz2@vsb.cz (S.R.); rostislav.kawulok@vsb.cz (R.K.); michal.sauer@vsb.cz (M.S.)

[2] TŘINECKÉ ŽELEZÁRNY, a. s., Průmyslová 1000, 73961 Třinec, Czech Republic; petra.turonova@trz.cz

\* Correspondence: ivo.schindler@vsb.cz

**Abstract:** The deformation behaviour of a coarse-grained as-cast medium-carbon steel, alloyed with 1.2% Mn, 0.8% Cr and 0.2% Mo, was studied by uniaxial compression tests for the strain rates of 0.02 s$^{-1}$–20 s$^{-1}$ in the unusually wide range of temperatures (650–1280 °C), i.e., in various phase regions including the region with predominant bainite content (up to the temperature of 757 °C). At temperatures above 820 °C, the structure was fully austenitic. The hot deformation activation energies of 648 kJ·mol$^{-1}$ and 364 kJ·mol$^{-1}$ have been calculated for the temperatures ≤770 °C and ≥770 °C, respectively. This corresponds to the significant increase of flow stress in the low-temperature bainitic region. Unique information on the hot deformation behaviour of bainite was obtained. The shape of the stress-strain curves was influenced by the dynamic recrystallization of ferrite or austenite. Dynamically recrystallized austenitic grains were strongly coarsened with decreasing strain rate and growing temperature. For the austenitic region, the relationship between the peak strain and the Zener–Hollomon parameter has been derived, and the phenomenological constitutive model describing the flow stress depending on temperature, true strain rate and true strain was developed. The model can be used to predict the forming forces in the seamless tubes production of the given steel.

**Keywords:** low-alloy steel; phase composition; hot flow curves; activation energy; dynamic recrystallization; grain size

## 1. Introduction

Medium-carbon steel low alloyed with Mn, Cr and Mo is designed for the production of seamless casings, pipe couplings and socket bars according to the API Specification 5CT, for grade N80Q—i.e., in the hardened and tempered state. In this case, yield stress in the range from 552 MPa to 758 MPa and a minimum ultimate tensile strength of 689 MPa are required. In a seamless tube mill, this material is commonly processed under relatively high temperatures. However, for the purposes of computer simulations of given forming processes, it is advantageous to have information about the deformation behaviour of this steel even at low temperatures, which may occur locally in the cases of uneven cooling of the rolled semi-finished product, e.g., in connection with low temperature rolling. The aim of the experiments was to obtain information about the flow stress and hot deformation activation energy of the given steel in a wide range of conditions as well as to compare the deformation behaviour between the high-temperature region of austenite and the low-temperature region associated with the occurrence of other structural components, especially bainite. The novelty of the works is that the flow

stress of the Mn–Cr–Mo low alloy steel was for the first time thoroughly investigated and compared in the extremely wide temperature range of 650–1280 °C, i.e., also in the bainitic phase region.

Bainitic steels are a frequent object of interest for researchers, often from other disciplines (study of phase transformations, heat treatment, transformation-induced plasticity, stress corrosion, etc.). For 55SiMnMo steel, the stress–strain relationship based on the dislocation density variation was described in the austenite region (above 950 °C), considering the influence of the dynamic softening mechanisms (recovery and recrystallization) [1]. For Mn–Mo–B bainitic steel and flow stress at temperature from 900 °C to 1200 °C, a constitutive equation has been established on the basis of a dislocation theory, work hardening and dynamic recovery theory, as well as the softening mechanisms of dynamic recrystallization [2]. The effects of strain rate on flow stress during ausforming in two bainitic steels were investigated in [3]. For medium-carbon steel, flow stress increased as expected with strain as well as strain rate due to the work hardening. However, flow stress behaviour of low-carbon steel showed different characteristics. The maximum stress appeared at the lowest strain rate of 0.001 s$^{-1}$. Microstructure observation confirmed that a bainitic transformation induced by plastic deformation has a dominant effect on the flow stress behaviour of this steel at strain rates of 0.001–1.0 s$^{-1}$, while flow stress at the strain rates of 0.1–5.0 s$^{-1}$ was influenced by the work hardening. Deformation mechanisms of ferrite/bainite multi-phase steels were investigated at room temperature in [4]. The effect of volume fraction of bainite on yield ratio, strain hardening exponent and uniform elongation was explained by the division and strain hardening capability of the stages in the modified Crussard–Jaoul analysis [5].

Knowledge of the apparent activation energy $Q$ [J·mol$^{-1}$] in hot forming for the given material enables, among others, to predict the maximum flow stress (i.e., the peak stress $\sigma_p$ [MPa]) value readily at the given temperature $T$ [K, °C] and strain rate $\dot{e}$ [s$^{-1}$]—see, e.g., [6,7]. The $Q$ value is ideally the material constant depending only on the chemical composition and microstructure of the particular material. For simplification, it is possible to omit the experimentally verified fact that the $Q$ value also depends on the amount of strain [8–14] or even on the combination of the strain rate and temperature (see, e.g., the apparent activation energy map for Ni-containing Fe–Mn–Al–C lightweight duplex steel in [15]).

The value of hot deformation activation energy is used in the rapid prediction of maximum flow stress under given conditions, or in physically based models describing hot flow stress, including the effect of dynamic softening. Numerous types of hot flow stress models have been derived and applied in the last few decades. Because of the complicated flow stress course under the hot deformation conditions, considerable number of the derived models is intended to describe just a specific part of the experimental hot flow curves—generally a strain range before or after the peak point. For example, the Cingara–McQueen model [16] and Solhjoo's models [17–20] belong among the models proposed to describe the before-peak range. A model of Ebrahimi et al. [21] or a model derived from the theory of Johnson–Mehl–Avrami–Kolmogorov (JMAK) [22] are intended to describe the after-peak range. On the other hand, some models are utilized to describe also an entire strain range, e.g., models utilizing the Garofalo's relation [23,24] or Johnson–Cook model [25]. The models often include a description of the peak-point coordinates (i.e., peak strain and peak stress) and steady-state stress coordinate. In order to describe these coordinates, the well-known Zener–Hollomon parameter $Z$ [s$^{-1}$] [26] is usually used, which requires the preliminary estimation of the $Q$ value—see, e.g., [9,27–29].

## 2. Experimental Material and Procedures

Samples were taken from a continuously cast semi-finished product of circular cross-section with a diameter of 410 mm, approximately 50 mm below the surface. The steel was melted and cast in TŘINECKÉ ŽELEZÁRNY, a. s. (Třinec, Czech Republic). In addition to the majority iron, the investigated steel also contained the following significant chemical elements: 0.29 C–1.20 Mn–0.27 Si–0.015 P–0.004 S–0.79 Cr–0.21 Mo–0.028 Al–0.0093 N (all in wt.%). Its initial structure was formed mainly by bainite with a network of ferrite and pearlite along the boundaries of the prior

and very coarse austenitic grains—see Figures 1 and 2. Metallographic samples were etched with 4% Nital solution (a mixture of nitric acid and ethanol) and observed on the Olympus GX51 inverted metallurgical microscope (Olympus Corporation, Tokyo, Japan). A scanning electron microscope JEOL JSM-6490LV (JEOL Ltd., Tokyo, Japan) and the energy-dispersive X-ray spectrometer (SEM-EDS) were used to obtain an element distribution map. Samples for SEM analysis were etched at room temperature for 5 s in a solution composed of 100 mL of ethanol, 10 mL of hydrochloric acid, 6 mL of acetic acid and 6 g of picric acid. SEM analyses were performed in secondary electron imaging (SEI) mode.

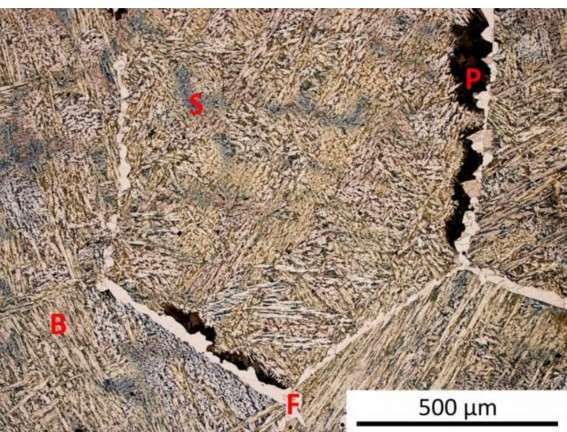

**Figure 1.** Microstructure of the investigated steel with bainite (**B**), ferrite (**F**) and perlite (**P**)—initial state; dendritic segregations (**S**) are grey (light microscopy).

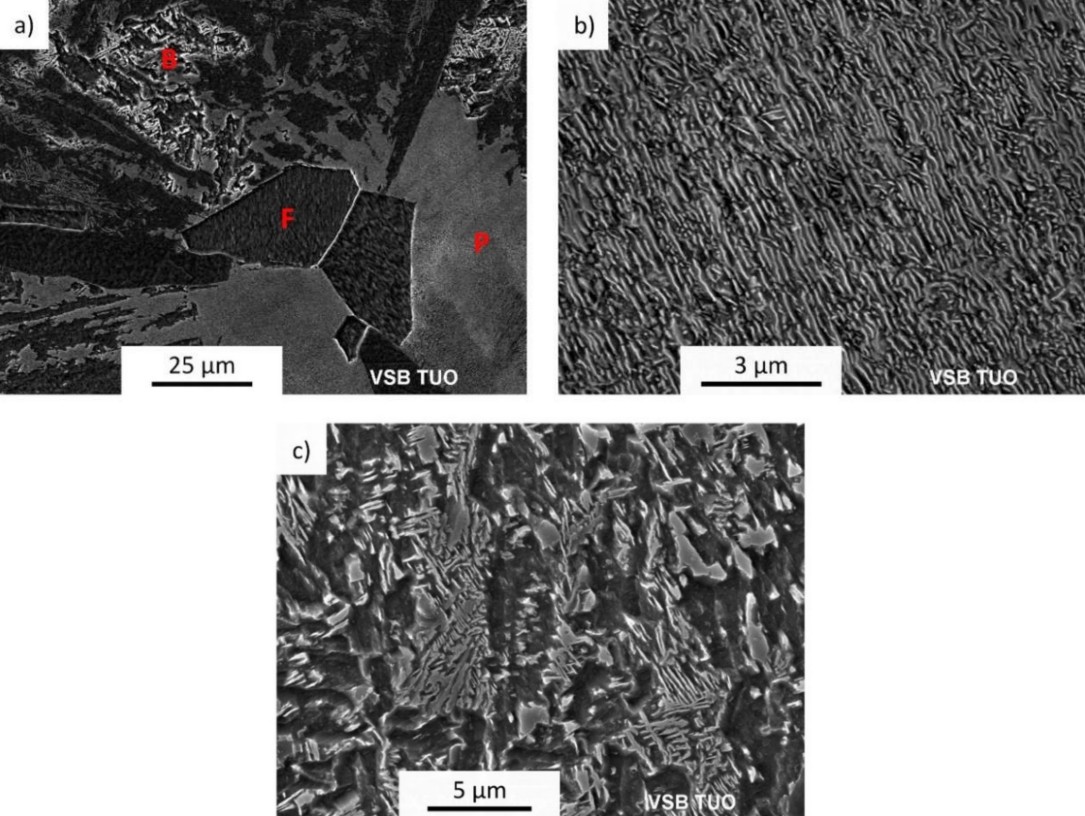

**Figure 2.** Initial microstructure—scanning electron microscopy (SEM) analysis. (**a**) Region of the prior austenitic grain boundary with ferrite (**F**), pearlite (**P**) and bainite (**B**); (**b**) pearlite detail; (**c**) bainite detail.

Quantitative structure analysis in QuickPHOTO Industrial 3.2 software (PROMICRA, s.r.o., Prague, Czech Republic) determined that the as-cast structure contained only about 1.7% ferrite and 0.7% pearlite at the boundaries of the prior austenitic grains. It has been found by the SEM-EDS analysis that dendritic segregations are associated with increased Mn and Cr content—see the lighter areas in Figure 3. A higher tendency to segregations can be expected for Mn, Cr and Mo in the studied steel. Nevertheless, Mo content in the steel (0.21%) is lower than Mn and Cr content (1.20% and 0.79%), which explains that only segregations of Mn and Cr were revealed by the SEM-EDS analysis.

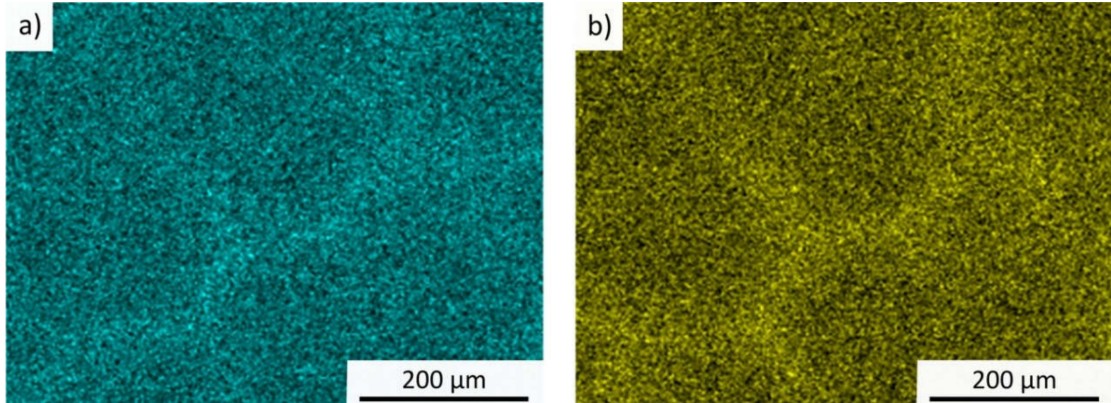

**Figure 3.** Results of X-ray elemental mapping for initial state of the material (K-alpha lines). (**a**) Manganese segregations; (**b**) chromium segregations.

Hot flow curves were studied in a such temperature range that the area of the existence of austenite and partly also the area of the coexistence of bainite and other structural components were covered. The highest temperature was chosen in accordance with the heating temperature of the steel before piercing in the rolling mill of seamless tubes, i.e., 1280 °C. Other temperatures were chosen based on the results of the dilatometric test. The DSI 39112 Scanning Non-Contact Optical Dilatometer System with a green LED technology (i.e., part of the Gleeble 3800 simulator—Dynamic Systems Inc., Poestenkill, NY, USA) was used for the dilatation measuring of the 6 mm diameter cylindrical sample heated by a rate of 0.167 °C·s$^{-1}$. The analysis of the dilatation curve was performed using the numerical derivative in the OriginPro software 9 (OriginLab Corporation, Northampton, MA, USA).

Cylindrical samples with a diameter of 10 mm and a height of 15 mm were produced from the supplied cuttings. On the Hydrawedge II mobile conversion unit of the Gleeble 3800 hot deformation simulator, the samples were subjected to uniaxial compression with the height reduction corresponding to a maximum true strain of $e = 1.0$. Tantalum foils and a special high-temperature lubricant were used to reduce a friction between the faces of the cylindrical sample and the anvils. The samples were heated directly to the testing temperature (i.e., 650–1280 °C). The individual testing temperatures were chosen to maintain approximately the equal $(1/T)$ intervals, where $T$ is the Kelvin temperature in this case. This procedure is advantageous in terms of accuracy in the subsequent activation energy calculations. The holding time of 300 s was followed by a deformation at a nominal true strain rate of 0.02 s$^{-1}$–0.2 s$^{-1}$–2.0 s$^{-1}$–20 s$^{-1}$. The true stress–true strain curves were automatically calculated from the measured and computer-registered stroke and force values when using the internal Gleeble algorithm. The calculation included compensation for the barreling effect at compression.

Selected samples were quenched with 4 water jets immediately after heating or deformation. By this way fixed structure was subsequently subjected to the metallographic analyses in the central area of the sample. In the case of the visualization of the prior austenitic grain boundaries, a solution of 50 g of picric acid and 5 g of ferric chloride in 100 mL of distilled water was used as an etchant. The quenched samples were pre-heated to 40 °C, and the etching time was chosen differently based on the partial results. The lineal intercept technique for measuring the grain size by the QuickPHOTO

Industrial 3.2 software was used, and only the grains with clearly drawn boundaries were taken into account. Thus, at least 105 grains were measured for each sample.

## 3. Results

### 3.1. Phase Regions

The determination of $A_{c1}$ and $A_{c3}$ temperatures (corresponding to the start and finish of the phase transformation to austenite during the heating) is clear from Figure 4. The structure is formed by austenite at temperatures above the $A_{c3}$ = 820 °C; by bainite, ferrite and pearlite at temperatures below the $A_{c1}$ = 757 °C; and by austenite and ferrite at temperatures of 757–820 °C. The selection of deformation temperatures reflected these phase regions in the following tests: 650 °C and 710 °C (bainite, ferrite and pearlite), 770 °C (austenite and ferrite), 840–930–1030–1140–1280 °C (austenite).

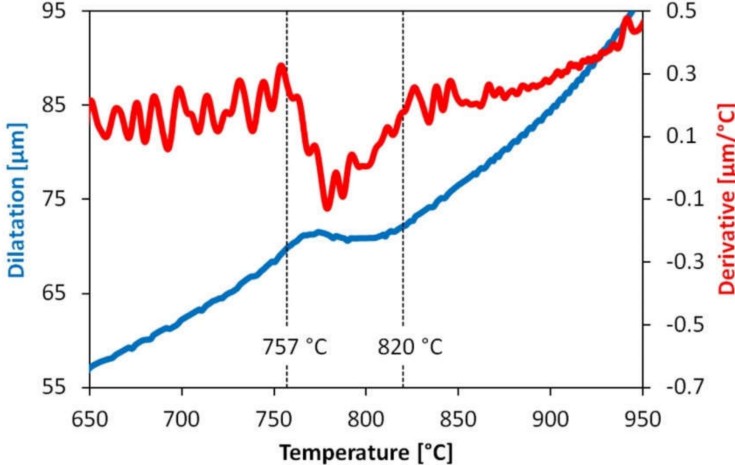

**Figure 4.** Analysis of the dilatation curve in the area of phase transformations, based on the numerical derivative of the dilatation-temperature data (a heating rate of 0.167 °C·s$^{-1}$).

### 3.2. Initial Microstructures after Heating

The structure of the samples heated to a temperature lower than $A_{c1}$ and then quenched does not differ in any way from the structure in the initial state—compare Figures 1 and 5.

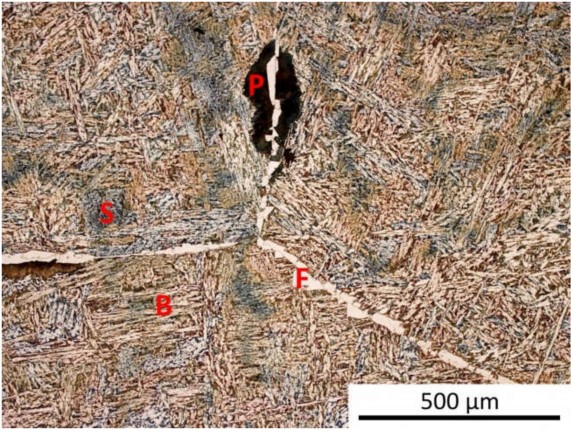

**Figure 5.** Structure of the sample quenched from a heating temperature of 650 °C—bainite (**B**), ferrite (**F**) and perlite (**P**); dendritic segregations (**S**) are grey (light microscopy).

The absence of pearlite in the respective sample confirms that the heating temperature of 770 °C falls between the $A_{c1}$ and $A_{c3}$ temperatures. Along the boundaries of the prior austenitic grains,

only ferrite occurs in a minimal amount of about 1.3%. Bainite predominates in the structure and martensite also occurs in the areas of segregation—see Figures 6 and 7.

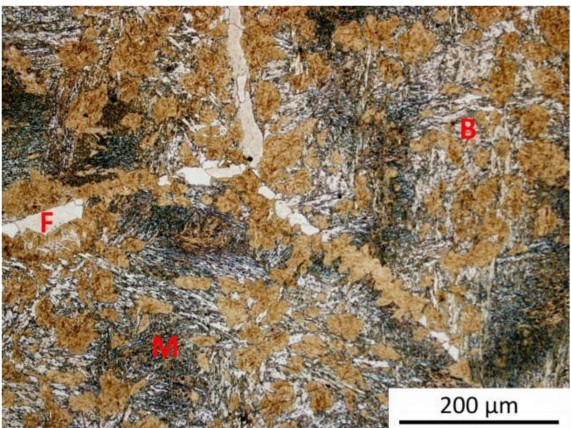

**Figure 6.** Structure of the sample quenched from a heating temperature of 770 °C; light microscopy.

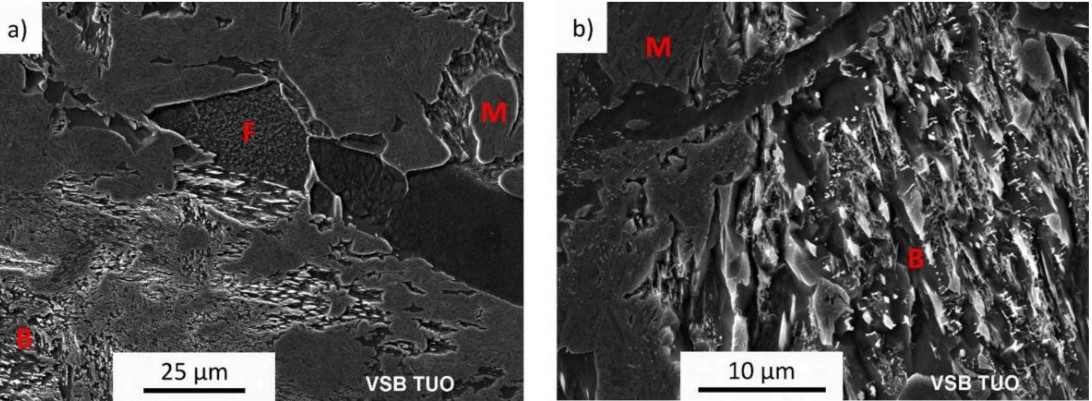

**Figure 7.** SEM analysis of a sample quenched from a heating temperature of 770 °C. (**a**) Three structure constituents—bainite (**B**), ferrite (**F**) and martensite (**M**); (**b**) close-up view of bainite and martensite.

During heating to the temperatures of 840–1280 °C (i.e., above the $A_{c3}$) the ferrite has already been transformed into austenite and the resulting structure is after the quenching always formed exclusively by martensite—see Figure 8 for example. It can be deduced from Figure 8a that pronounced segregations (S) were present in the structure, which remained nearly intact during etching. A detailed view of lath martensite is shown in Figure 8b.

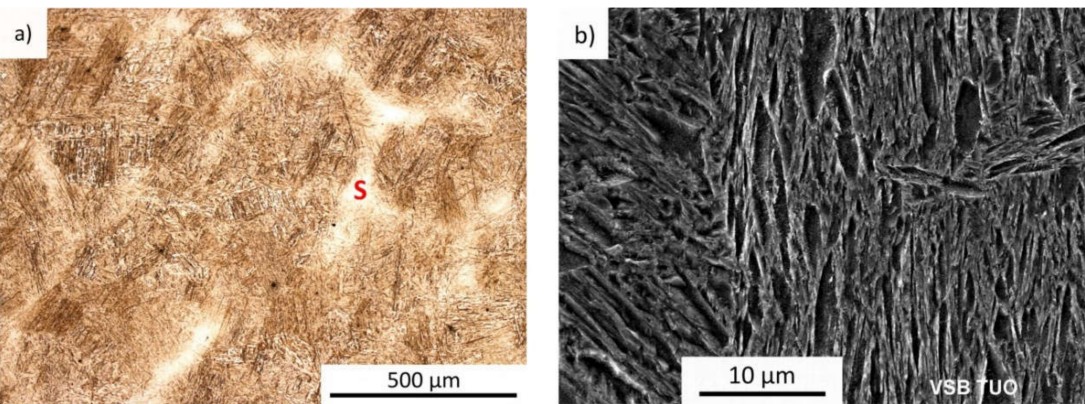

**Figure 8.** Microstructure of the sample quenched from a heating temperature of 1140 °C. (**a**) Light microscopy—segregations in bright areas, marked S; (**b**) SEM—close-up view of lath martensite.

### 3.3. Hot Flow Curves

The experimental true stress–true strain curves were smoothed using the OriginPro software. As documented in Figure 9, this procedure was especially important in the case of the highest strain rate level; the green squares correspond to the combination of low temperature and low strain rate, the ochre squares correspond to the test performed at medium temperature and high strain rate.

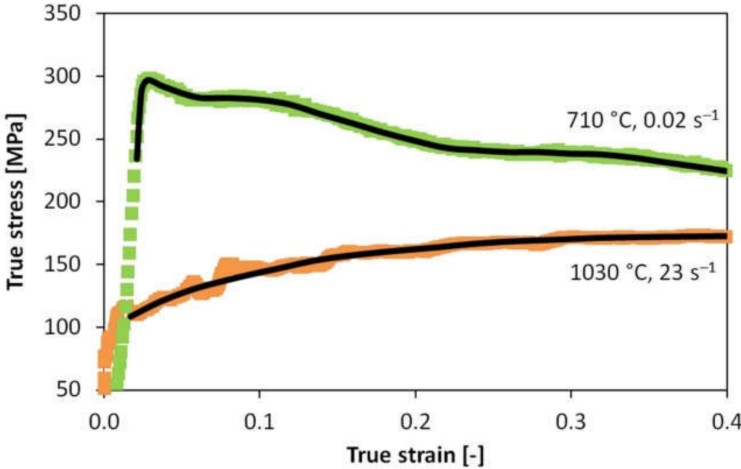

**Figure 9.** Example of flow curves smoothing (coloured points correspond to experimental values, black lines are the result of smoothing and interpolation).

For each curve, both coordinates of the peak point (i.e., stress $\sigma_p$ [MPa] and strain $e_p$ [–]) were located and the stress values $\sigma$ [MPa] for the strain values of $e$ = 0.02; 0.04 ... 0.98; 1.00 were determined. These interpolated stress values are plotted in Figures 10–14. The Figures 10 and 11 compare the flow curves obtained for the lowest and highest strain rate level. The Figures 12–14 demonstrate the effect of strain rate on the flow curves at selected temperature levels, i.e., 1280 °C, 840 °C and 650 °C. In this case, the stated strain rate values are the mean values calculated for each test from the registered data.

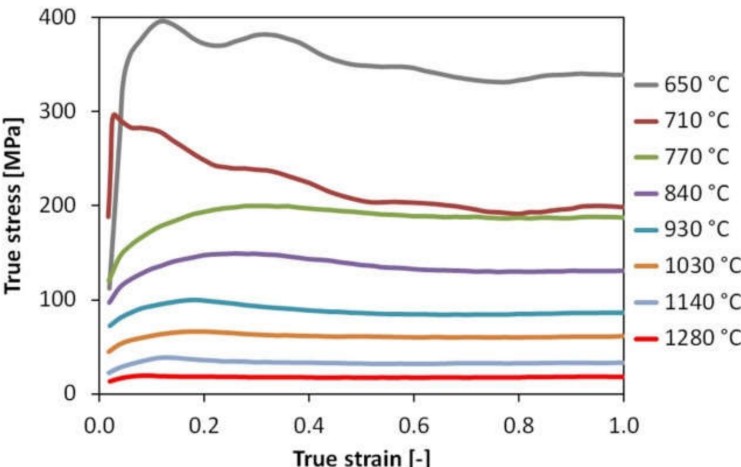

**Figure 10.** Influence of temperature on flow curves at a nominal strain rate of 0.02 s$^{-1}$.

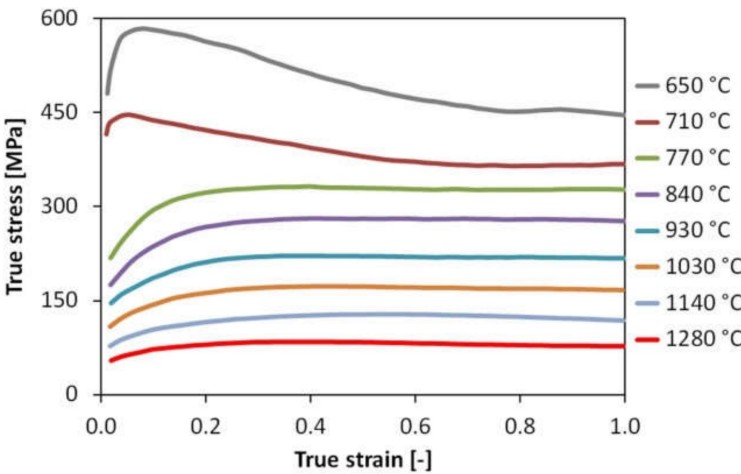

**Figure 11.** Influence of temperature on flow curves at a nominal strain rate of 20 s$^{-1}$.

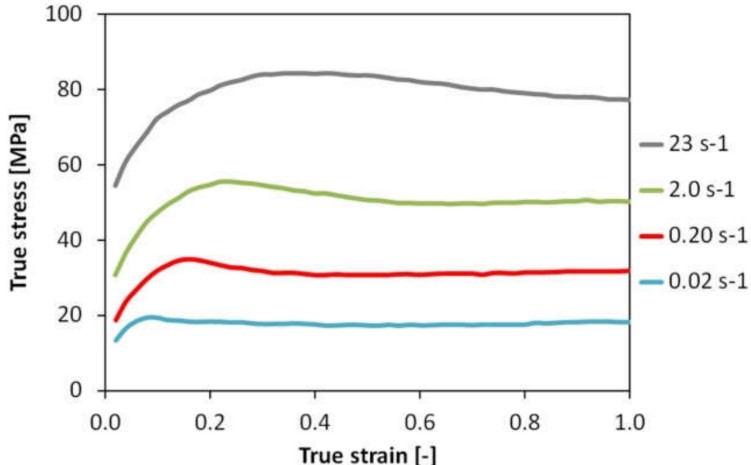

**Figure 12.** Influence of strain rate on flow curves at a nominal temperature of 1280 °C.

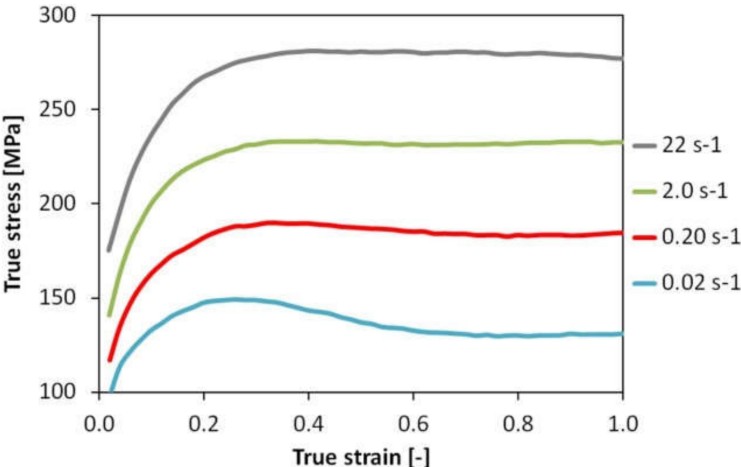

**Figure 13.** Influence of strain rate on flow curves at a nominal temperature of 840 °C.

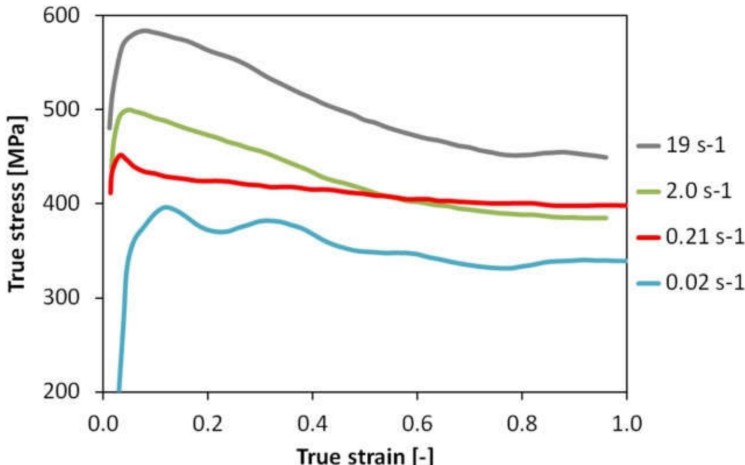

**Figure 14.** Influence of strain rate on flow curves at a nominal temperature of 650 °C.

All curves have a shape corresponding to the combination of strain hardening and dynamic recrystallization. However, at high strain rates in particular, the curves tend to be quite flat, which makes it difficult to reliably and accurately locate the $e_p$ value. With decreasing temperature and increasing strain rate, both peak stress and strain values generally increase; this is surprisingly also applied at a temperature of 770 °C, where a different deformation behaviour against the pure austenite at higher temperatures was already assumed. The character of the curves is sharply changed at the temperatures of 710 °C and 650 °C, i.e., in the case of a structure consisting mainly of bainite. At low strain rate values, the curves are wavy, as if there were a significant cyclic course of dynamic recrystallization. The similar cyclic behaviour was observed, for example, in the plain carbon steels with respectively 0.06% and 0.5% C [30] and in the medium Mn steel [31]—in austenite and preferably at very low strain rates. The stress peaks are much more pronounced than in the austenite region, the $e_p$ values are low and practically independent of strain rate. The shape of the stress-strain curves in the bainite region is similar to that of slow deformation of ferrite in the Ti+Nb stabilized interstitial free steel, but only at high temperatures (at least 800 °C) [32].

## 4. Mathematical Processing of Experimental Data and Discussion

### 4.1. Hot Deformation Activation Energy

The apparent activation energy $Q$ [J·mol$^{-1}$] in hot forming is considered as a very important material constant, used primarily for the calculation of the Zener–Hollomon parameter $Z$ [s$^{-1}$] [26]:

$$Z = \dot{e} \cdot \exp\left(\frac{Q}{R \cdot T}\right)$$ (1)

where $\dot{e}$ [s$^{-1}$] is true strain rate, $T$ [K] is temperature and $R = 8.314$ J·mol$^{-1}$·K$^{-1}$ is the gas constant; parameter $Z$ represents the temperature-compensated strain rate.

The hyperbolic law in Arrhenius-type equation is commonly used for determination of the activation energy value [33]:

$$\dot{e} = C \cdot \exp\left(\frac{-Q}{R \cdot T}\right) \cdot \left[\sinh\left(\alpha \cdot \sigma_p\right)\right]^n$$ (2)

where $C$ [s$^{-1}$], $n$ [-] and $\alpha$ [MPa$^{-1}$] are other material constants. In most cases, this relationship is solved by a simple graphic method based on the repeatedly used linear regression [34]. The specificity of the

hyperbolic function is used in this calculation that simplifies the Equation (2) for low stress values (i.e., $\alpha \cdot \sigma_p < 0.8$) into the form of the power law:

$$\dot{e} = C_1 \cdot \exp\left(\frac{-Q}{R \cdot T}\right) \cdot \sigma_p{}^n \tag{3}$$

and, vice versa, for high stress values (i.e., $\alpha \cdot \sigma_p > 1.2$) into the form of the exponential law:

$$\dot{e} = C_2 \cdot \exp\left(\frac{-Q}{R \cdot T}\right) \cdot \exp\left(\beta \cdot \sigma_p\right) \tag{4}$$

where $C_1$, $C_2$ and $\beta$ are the material constants. The $\alpha$ value in Equation (2) is given by the formula $\alpha = \beta/n$. For a chosen high-temperature level (with low stress values), the constant $n$ can be determined by the linear regression of the experimental data in the coordinates $\ln \dot{e} \sim \ln \sigma_p$. For a chosen low-temperature level (with high stress values), the constant $\beta$ can be obtained by the linear regression in the coordinates $\ln \dot{e} \sim \sigma_p$. After the calculation of the $\alpha$ quantity, the constants $Q$ and $C$ in Equation (2) can be calculated by the final linear regression of all experimental data plotted in the coordinates $\ln \dot{e} - n \cdot \ln(\sinh(\alpha \cdot \sigma_p)) \sim T^{-1}$. Unfortunately, such an estimate of the constants $n$ and $\beta$ is markedly influenced by the selection of the individual temperature levels. This shortcoming can be eliminated by the application of the specially developed software ENERGY 4.0 (VŠB—TU Ostrava, Czech Republic) [35], which uses the aforementioned values of $n$ and $\beta$ only as the preliminary estimate of parameters for final refining via the non-linear regression analysis of all data corresponding to Equation (2).

Calculation of the hot deformation activation energy from experimental peak stress values on the basis of Equation (2) is the approved method that was applied for different types of metallic materials, recently for example for various steel grades [7,36–39], intermetallic compounds [6,40–43], alloys based on aluminium [12,44–46], copper [47,48], titanium [49,50], magnesium [51,52], cobalt [53,54], etc.

Due to the existence of different structural states in the high-temperature region (austenite) and low-temperature region (austenite + ferrite or bainite + ferrite + perlite), it was not possible to uniformly calculate the activation energy value based on Equation (2) in the whole deformation temperature range (i.e., 650–1280 °C). Therefore, an attempt was made to find the temperature separating these two temperature ranges in repeated calculations. Surprisingly, it was possible to determine with a good accuracy the value of the activation energy for the temperature range of 770–1280 °C, respectively for the 650–770 °C, although the temperature of 770 °C should already correspond to a two-phase structure (but only with a small share of ferrite). Using these $Q$ values as well as other regression-calculated material constants and after adjusting the Equation (2), the peak stress of the investigated steel could be described by the following relations:

- for a temperature range of 770–1280 °C ($Q = 364$ kJ·mol$^{-1}$):

$$\sigma_p = \frac{1}{0.0108} \cdot \operatorname{arcsinh} \sqrt[4.87]{\frac{Z}{3.46 \cdot 10^{13}}} \tag{5}$$

- for a temperature range of 650–770 °C ($Q = 648$ kJ·mol$^{-1}$):

$$\sigma_p = \frac{1}{0.0029} \cdot \operatorname{arcsinh} \sqrt[11.8]{\frac{Z}{1.84 \cdot 10^{33}}} \tag{6}$$

The values of apparent activation energy in hot forming calculated by the applied method are average values for both temperature ranges, and the method of their calculation, as well as the choice of the range of deformation temperatures, could influence them to a certain extent [15,35,37]. In this concept, the $Q$ value is one of the material constants in Equation (2), and the determination of their optimal combination depends on the experimental conditions and the procedure of regression analysis of the data obtained. Anyway, it is clear that after falling below the temperature of 770 °C,

the deformation behaviour of the investigated steel is changed significantly and the stress–strain curves acquire a shape with a very pronounced peak, achieved with relatively small strains. The increase in flow stress compared to the austenitic region is due to the majority of bainite in the structure—see Figures 1 and 5 for example. At temperatures below the $A_{c3}$, there was a sufficient content of softer ferrite in the structure with a high number of potential slip systems (48 systems types {1 1 0} ⟨1 1 1⟩, {1 1 2} ⟨1 1 1⟩ and {1 2 3} ⟨1 1 1⟩ for the body-centred cubic structure of ferrite vs. 12 systems {1 1 1} ⟨1 1 0⟩ for the face-centred cubic structure of austenite [55,56]), this would lead to a decrease in flow stress. This phenomenon is best observed in the case of non-alloy steels with very low carbon content, such as interstitial free (IF) steel with 0.004% C and 0.072% Ti—see Figure 15. The mean flow stress values were calculated from the measured rolling forces and used to derive the equations published in [57], on the basis of which the curves in Figure 15 were plotted for a true strain of 0.4 and true strain rate of 5 s⁻¹.

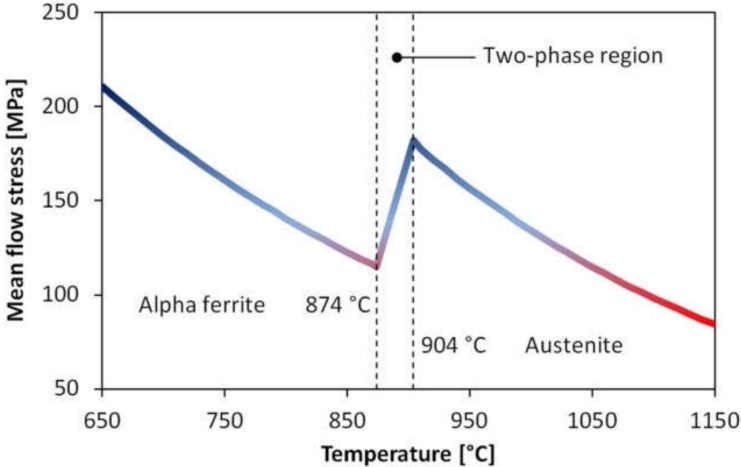

**Figure 15.** Influence of phase composition on mean flow stress of the Ti–IF steel.

Analogous results were published in [58] for the low-carbon steel. The flow stress of the austenite increases with a decreasing deformation temperature down to about 875 °C. At temperatures lower than 825 °C, the flow stress in the ferrite range increases more rapidly with a decreasing temperature than observed for the austenite. In the austenite+ferrite region, the flow stress falls with a decreasing temperature. The $Q$ value of austenite and ferrite was determined to be about 322 kJ·mol⁻¹ and 219 kJ·mol⁻¹, respectively. Similarly, hot deformation activation energy of the IF steel decreases from 324 kJ·mol⁻¹ to 285 kJ·mol⁻¹ in ferritic phase region and 365 kJ·mol⁻¹ to 342 kJ·mol⁻¹ in austenite phase region with a rise in the true strain from 0.05 to 0.60 [32].

The effect of δ-ferrite can be similar as observed for example in a Fe–26Mn–6.2Al–0.05C steel [59] with duplex matrix composed of austenite and δ-ferrite. As the deformation temperature increased from 800 °C to 1050 °C, the volume fraction of δ-ferrite increased from 29% to 56%. The flow stress is mostly concentrated in δ-ferrite during the early stages of plastic deformation. Therefore, dynamic recovery and dynamic recrystallization occur earlier in δ-ferrite than in austenite. This was reflected in the calculation of hot deformation activation energy values. The average $Q$ values at high (>900 °C) and low (≤900 °C) temperatures were determined as 395 kJ·mol⁻¹ and 466 kJ·mol⁻¹, respectively.

A similarly large difference in $Q$ values for the low-temperature and high-temperature regions, as in the case of the investigated low-alloy Mn–Cr–Mo steel, was also observed for a multiphase Fe–11Mn–10Al–0.9C lightweight steel: 601 kJ·mol⁻¹ for the deformation temperature ranging from 800 °C to 900 °C and 356 kJ·mol⁻¹ for the temperature ranging from 950 °C to 1100 °C [38]. However, the reasons for this are quite different. The inhibiting effect of dynamic precipitation of κ-carbides on dynamic recrystallization led to the higher value of activation energy obtained at low temperatures.

A significant difference in hot deformation activation energy has been observed in diverse phase regions for titanium alloys. The hot deformation activation energy of SP-700 alloy in the $\alpha + \beta$ region was higher than in the single-phase $\beta$ region (305 kJ·mol$^{-1}$ vs. 165 kJ·mol$^{-1}$) due to the dynamic globularization of the lamellar $\alpha$ phase [60]. A similar increase in $Q$ value due to the formation of the $\alpha$ phase was observed for several titanium alloys: 495 kJ·mol$^{-1}$ in the $\alpha + \beta$ region vs. 196 kJ·mol$^{-1}$ in the $\beta$ region for a Cu-bearing antibacterial Ti-645 alloy [61], 453 kJ·mol$^{-1}$ vs. 280 kJ·mol$^{-1}$ for a Ti-55 high-temperature alloy [62], 617 kJ·mol$^{-1}$ vs. 149 kJ·mol$^{-1}$ for a TC8M-1 alloy [63], 589 kJ·mol$^{-1}$ vs. 226 kJ·mol$^{-1}$ a TA15 alloy [64], 292 kJ·mol$^{-1}$ vs. 180 kJ·mol$^{-1}$ for a Ti–5Al–5Mo–5V–1Cr–1Fe alloy [65], 275 kJ·mol$^{-1}$ vs. 148 kJ·mol$^{-1}$ for a Ti–5Al–5Mo–5V–3Cr–1Zr alloy [66].

Alloying additives, other structural components as well as various deformation and softening mechanisms generally complicate the situation with hot deformation behaviour. For example, shape of the flow curves of the magnesium alloy AZ31 differs markedly at low and high values of the Zener–Hollomon parameter [67,68]. The reasons for these differences do not lie in phase transformations, but in the strong effect of twinning at initial stages of deformation in the case of high Z values. At low values of the temperature-compensated strain rate, the influence of dynamic recrystallization is dominant.

## 4.2. Peak Stress

The experimental values and according to Equations (5) and (6) calculated values of the $\sigma_p$ are graphically compared in Figure 16. The solid and empty triangles correspond to the temperature ranges of 770–1280 °C and 650–710 °C, respectively; these points are plotted taking into account the activation energy $Q$ = 364 kJ·mol$^{-1}$ when calculating the Zener–Hollomon parameter. For the temperature range of 650–770 °C it is suitable to choose the activation energy $Q$ = 648 kJ·mol$^{-1}$—see the solid diamonds in Figure 16. The effect of the $Q$ magnitude on the calculated values of the $Z$ parameter is remarkable. Table 1 lists all experimentally determined and according to Equations (5) and (6) calculated values of peak stress depending on the temperature and strain rate.

The accuracy of the mathematical description of the dependence $\sigma_p = f(Z)$ is very good in the whole range of the experimental conditions. If we consider the absolute deviations of the calculated data, an average value of 0.1 MPa with a standard deviation of 2.7 MPa corresponds to the high-temperature region and an average value of 0.1 MPa with a standard deviation of 7.2 MPa is associated with the low-temperature region. The relative deviations show values of 0.4% ± 2.9% for the high-temperature range and 0.1% ± 2.1% for the low-temperature range. The accuracy of the $\sigma_p$ values description for the boundary temperature of 770 °C is similar in the case of using Equations (5) and (6)—see Figure 17.

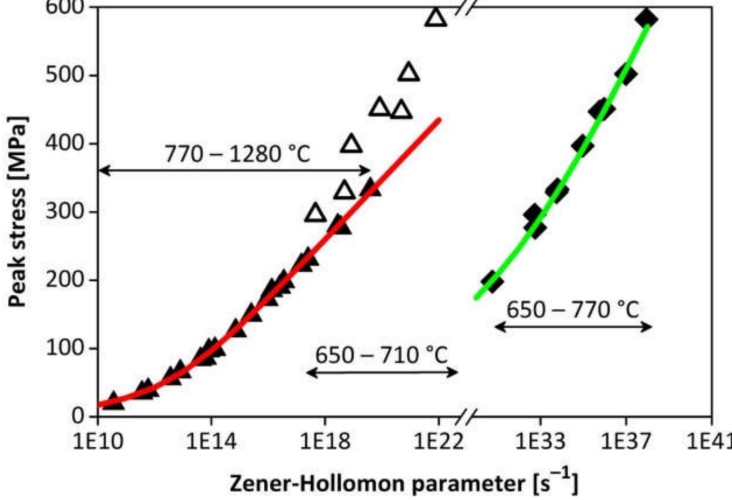

**Figure 16.** Comparison of the $\sigma_p$ values obtained experimentally (various points) and predicted according to Equation (5) (red line), or according to Equation (6) (green line).

**Table 1.** Comparison of the measured and predicted peak stress values for the high-temperature and low-temperature regions.

| $T$ [°C] | $\dot{e}$ [s$^{-1}$] | $Z$ [s$^{-1}$] | $\sigma_p$ [MPa] Experimental | $\sigma_p$ [MPa] Calculated |
|---|---|---|---|---|
| 1280 | 0.02 | $3.8 \times 10^{+29}$ | 20 | 22 |
| 1280 | 0.20 | $3.8 \times 10^{+30}$ | 35 | 35 |
| 1280 | 2.0 | $3.8 \times 10^{+31}$ | 56 | 55 |
| 1280 | 23 | $4.4 \times 10^{+32}$ | 84 | 84 |
| 1140 | 0.02 | $4.8 \times 10^{+32}$ | 39 | 39 |
| 1140 | 2.1 | $5.0 \times 10^{+34}$ | 86 | 90 |
| 1140 | 24 | $5.7 \times 10^{+35}$ | 126 | 127 |
| 1030 | 0.02 | $3.8 \times 10^{+35}$ | 66 | 64 |
| 1030 | 0.20 | $3.8 \times 10^{+36}$ | 97 | 93 |
| 1030 | 23 | $4.4 \times 10^{+38}$ | 172 | 173 |
| 930 | 0.02 | $4.8 \times 10^{+38}$ | 99 | 101 |
| 930 | 2.0 | $4.8 \times 10^{+40}$ | 185 | 179 |
| 930 | 22 | $5.2 \times 10^{+41}$ | 222 | 223 |
| 840 | 0.02 | $8.8 \times 10^{+41}$ | 149 | 149 |
| 840 | 0.20 | $8.8 \times 10^{+42}$ | 190 | 191 |
| 840 | 2.0 | $8.8 \times 10^{+43}$ | 231 | 234 |
| 840 | 22 | $9.6 \times 10^{+44}$ | 280 | 279 |
| 770 | 0.02 | $7.4 \times 10^{+44}$ | 198 | 197 |
| 770 | 2.0 | $7.4 \times 10^{+46}$ | 277 | 283 |
| 770 | 22 | $8.2 \times 10^{+47}$ | 333 | 329 |
| 770 | 0.02 | $7.4 \times 10^{+44}$ | 198 | 202 |
| 770 | 2.0 | $7.4 \times 10^{+46}$ | 277 | 283 |
| 770 | 22 | $8.2 \times 10^{+47}$ | 333 | 334 |
| 710 | 0.02 | $5.2 \times 10^{+47}$ | 296 | 282 |
| 710 | 0.21 | $5.4 \times 10^{+48}$ | 329 | 332 |
| 710 | 21 | $5.4 \times 10^{+50}$ | 447 | 441 |
| 650 | 0.02 | $8.4 \times 10^{+50}$ | 397 | 396 |
| 650 | 0.20 | $8.4 \times 10^{+51}$ | 451 | 453 |
| 650 | 2.1 | $8.8 \times 10^{+52}$ | 502 | 515 |
| 650 | 19 | $8.0 \times 10^{+53}$ | 582 | 575 |

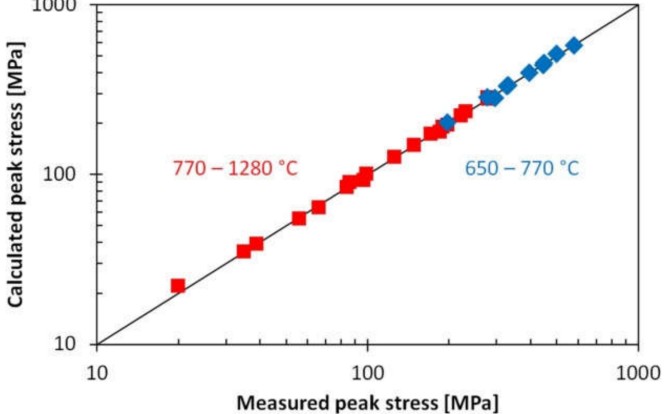

**Figure 17.** Accuracy of the values $\sigma_p$–values prediction according to Equation (5) (red points) and Equation (6) (blue points).

*4.3. Peak Strain*

The mathematical description of the peak strain was marked by the considerable variance of data in the high-temperature region and completely disappointing in the low-temperature region. In Figure 18, therefore, the parameter $Z$ calculated with the value $Q = 364$ kJ·mol$^{-1}$ was used for the plotting of all experimental values of $e_p$ (including the data corresponding to the temperatures of 650 °C and 710 °C). The common type of power function [69,70] was used for the regression analysis of the peak strain data in the temperature region from 770 °C to 1280 °C:

$$e_p = 0.0024 \cdot Z^{0.07} \tag{7}$$

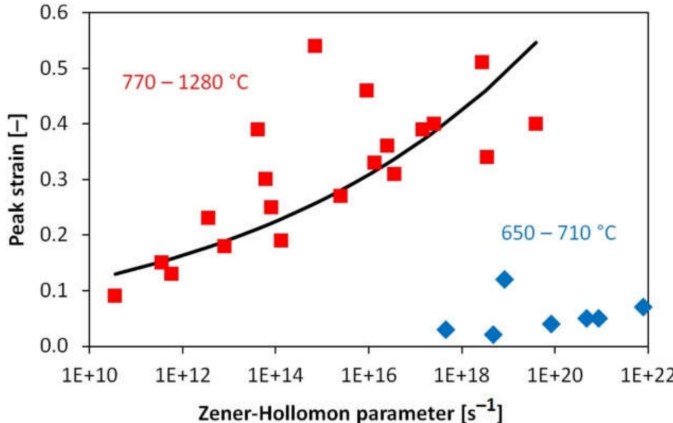

**Figure 18.** Experimental values of $e_p$ (coloured points) and their mathematical description in the high-temperature region by Equation (7) (black line).

Deformation behaviour of the bainitic rail steel with 0.22 C–1.84 Mn–0.91 Si–0.80% Cr–0.32% Mo was studied by the set of compression tests in the austenite region (temperature of 900–1200 °C, strain rate of 0.01 s$^{-1}$–10$^{-1}$) [71]. By analysing the published graphs, data on the experimentally determined peak stress coordinates can be obtained. The peak strain values ranged from 0.11 to 0.51, which corresponds very well to the values obtained (albeit in a slightly different range of experimental conditions) for the investigated Mn–Cr–Mo steel – see the red points in Figure 18. Unfortunately, a more accurate comparison is not possible because the necessary activation energy data are not available for Mn–Si–Cr–Mo rail steel. The data in Table 2 show that the Mn–Cr–Mo steel has a higher deformation resistance under comparable conditions with the exception of high temperatures.

**Table 2.** Comparison of the peak stress values for two bainitic steel grades.

| $T$ [°C] | $\dot{e}$ [s$^{-1}$] | $\sigma_p$ [MPa] | |
|---|---|---|---|
| | | Mn–Si–Cr–Mo Steel—Experimental [71] | Mn–Cr–Mo Steel—Calculated, Equation (5) |
| 900 | 0.1 | 117 | 143 |
| 1000 | 0.1 | 70 | 95 |
| 1100 | 0.1 | 53 | 62 |
| 1200 | 0.1 | 42 | 42 |
| 1000 | 0.01 | 44 | 65 |
| 1000 | 1.0 | 130 | 132 |
| 1000 | 10 | 157 | 172 |

*4.4. Austenite Grains after Dynamic Recrystallization*

To verify the mechanism of the dynamic softening of deformed austenite, selected quenched samples were subjected to the metallographic analysis in order to visualize the boundaries of the prior austenitic grains. The micrographs in Figure 19 document the common effect of higher heating temperature on the growth of austenitic grain.

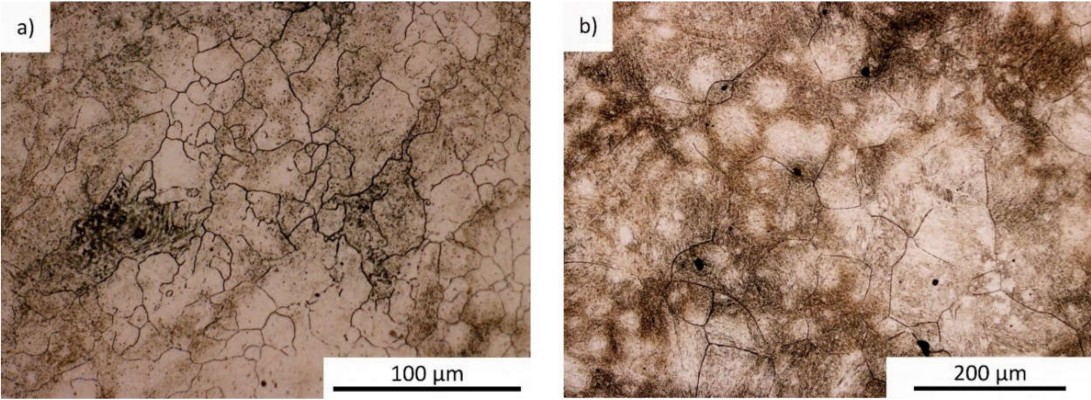

**Figure 19.** Prior austenitic grains after heating to different temperatures. (**a**) 840 °C; (**b**) 1280 °C.

Deformation below the $A_{c1}$ temperature led to the development of a heavily deformed anisotropic structure with the formations flattened by the plastic deformation, as shown in Figure 20.

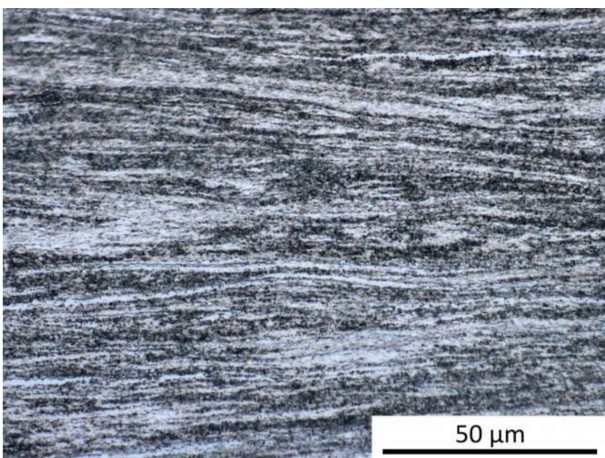

**Figure 20.** Banded structure of the sample deformed at a temperature of 650 °C (strain of 1.0, strain rate of 0.02 s$^{-1}$; the compression was set in the vertical direction of the image)—light microscopy.

A more detailed SEM analysis revealed that deformation at a temperature of 650 °C led to the dynamic recrystallization of bainitic or proeutectoid ferrite—see Figures 21 and 22. Equiaxed recrystallized grains have a size of about 0.5 µm–1.5 µm. In the sample tested at a low strain rate of 0.02 s$^{-1}$, flattened small ferrite grains can be found—due to deformation of the dynamically recrystallized grains.

None of the samples compressed at a temperature of at least 770 °C (i.e., with a structure formed predominantly or completely by austenite) showed the deformed prior grains after strain $e = 1.0$; all austenitic grains were dynamically recrystallized, but with very different sizes (see examples in Figure 23 for various deformation parameters).

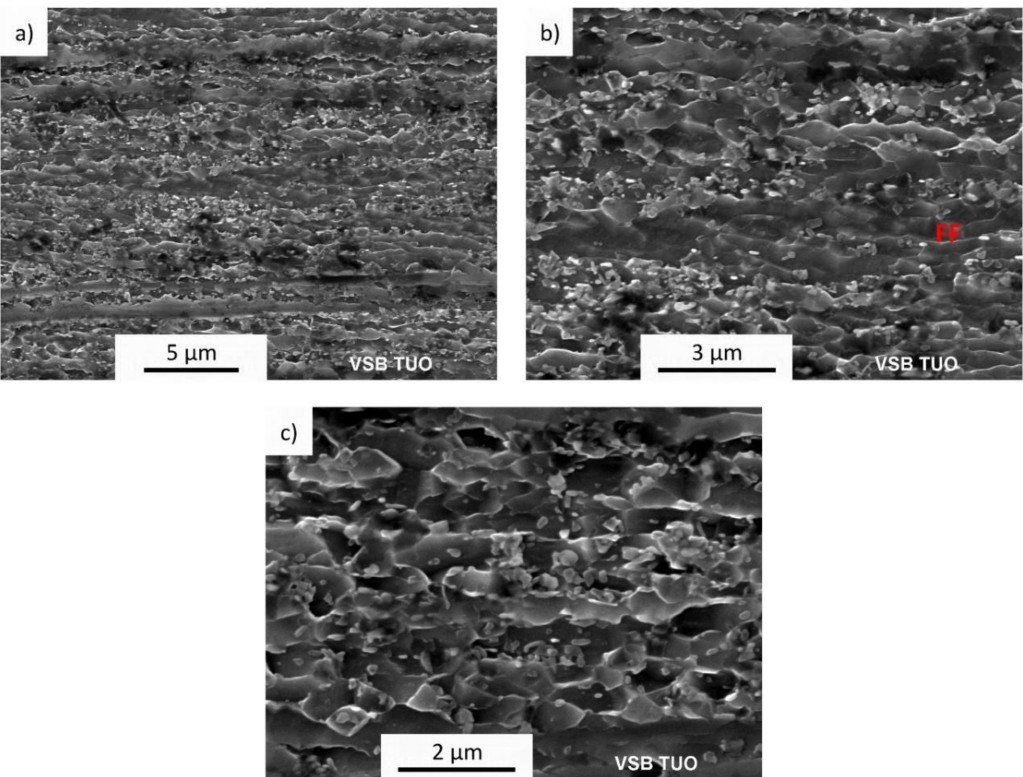

**Figure 21.** SEM analysis of the sample deformed at a temperature of 650 °C and a strain rate of 0.02 s$^{-1}$. (**a**) Banding as a result of the compression deformation; (**b**) flattened ferritic (FF) grains; (**c**) equiaxed ferritic grains and cementite (small bright particles).

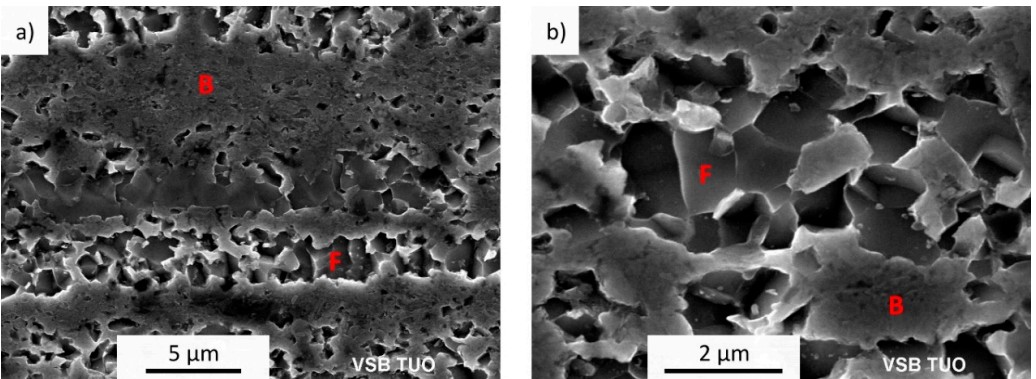

**Figure 22.** SEM analysis of the sample deformed at a temperature of 650 °C and a strain rate of 20 s$^{-1}$. (**a**) flattened bainitic agregates (**B**) and recrystallized ferritic grains (**F**); (**b**) detail of the equiaxed recrystallized grains of ferrite.

The austenite grain size values are compared in Table 3 and Figure 24. Confidence intervals were estimated for the significance level $\alpha = 0.05$.

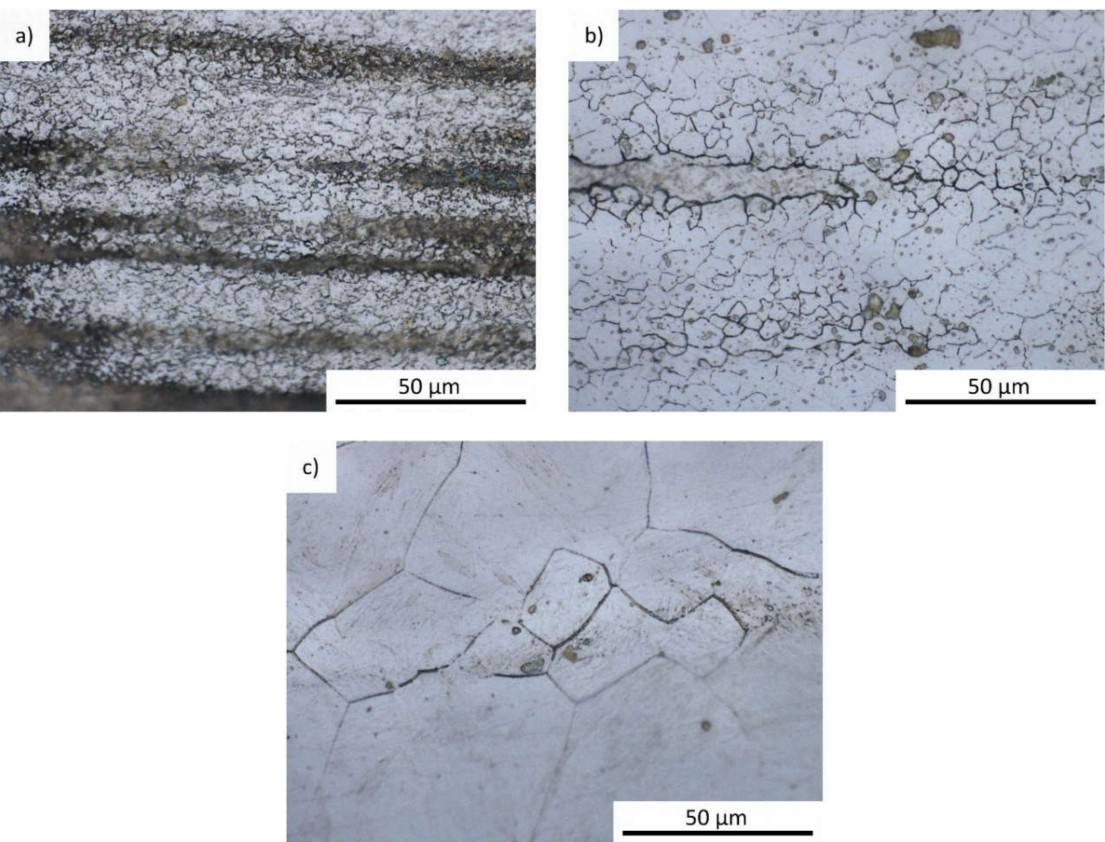

**Figure 23.** Prior austenitic grains in variously deformed samples. (**a**) temperature 770 °C, strain rate 0.02 s$^{-1}$; (**b**) temperature 930 °C, strain rate 20 s$^{-1}$; (**c**) temperature 1280 °C, strain rate 20 s$^{-1}$.

**Table 3.** Prior austenitic grain size after heating or hot deformation.

| Temperature [°C] | Mean Grain Size [µm] | Confidence Interval [µm] |
|---|---|---|
| | After heating | |
| 840 | 15.4 | ±1.6 |
| 1030 | 18.7 | ±1.9 |
| 1140 | 23.4 | ±2.5 |
| 1280 | 94.5 | ±8.4 |
| | After deformation (a strain rate of 0.02 s$^{-1}$) | |
| 770 | 3.0 | ±0.2 |
| 930 | 29.2 | ±3.0 |
| 1280 | 116.1 | ±5.2 |
| | After deformation (a strain rate of 20 s$^{-1}$) | |
| 770 | 2.4 | ±0.1 |
| 930 | 5.4 | ±0.3 |
| 1280 | 15.4 | ±2.0 |

During heating, the grain begins to significantly grow after the exceeding a temperature of about 1150 °C. Dynamically recrystallized grain is very fine in the case of a deformation temperature of 770 °C, but at higher temperatures it also significantly depends on the strain rate. Deformation at a strain rate of 20 s$^{-1}$ always led to a relative grain refinement. However, an inappropriate combination of low strain rate and high temperature results in the coarsening of dynamically recrystallized grains, the final size of which slightly exceeds even the size of the prior austenitic grains. This phenomenon was also observed in the case of dynamic recrystallization of 40Cr steel [72] when the microstructure

had a longer time for grain growth in the case of a low strain rate of 0.01 s$^{-1}$, and a higher strain rate was beneficial for the homogeneous fine-grain microstructure.

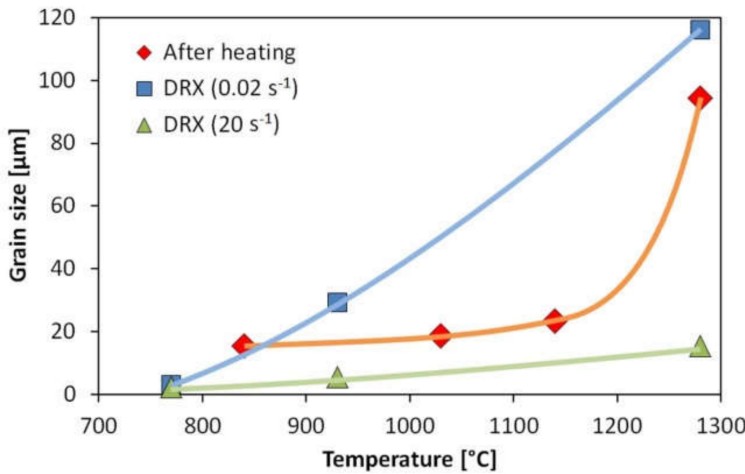

**Figure 24.** Influence of the processing parameters (heating vs. hot deformation) on austenitic grain size; DRX means dynamic recrystallization for two levels of the nominal strain rate.

Using the linear regression in Figure 25, it was possible to describe the dependence of the dynamically recrystallized grain size $D_{drx}$ [µm] on the Zener–Hollomon parameter (using the value $Q = 364$ kJ·mol$^{-1}$ for calculations) by the relationship:

$$D_{drx} = 10292 \cdot Z^{-0.194} \tag{8}$$

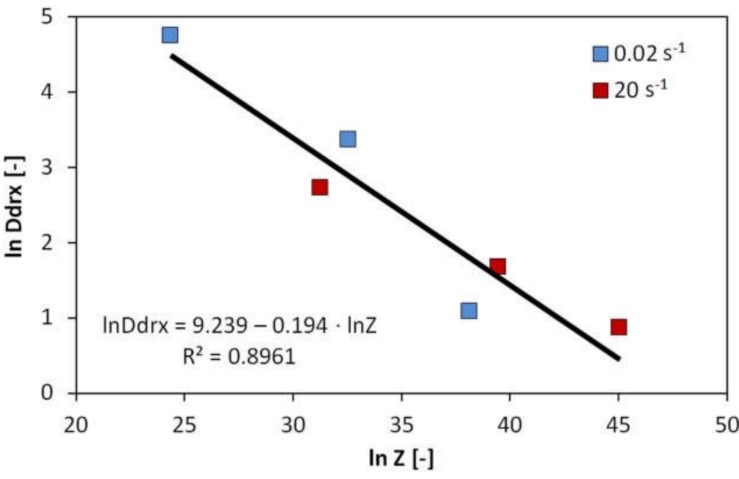

**Figure 25.** Graphical representation of the influence of deformation conditions on the size of dynamically recrystallized grain (strain $e = 1.0$).

A similar power relationship $D_{drx} = f(Z)$ was noted, for example, in the case of grade 40Cr steel, [72], grade 22MnB5 steel [73], grade 316LN austenitic stainless steel [74] and Cu–0.36Cr–0.03Zr alloy [75]. However, some other alloys exhibit rather different behaviour. For the medium-high-carbon high-silicon bainitic steel [76], two individual power functions had to be used for the description of the influence of temperature and strain rate, respectively. The respective dependencies varied in the case of different deformation temperature or strain rate values when testing the biomedical Co–Cr–W-based alloy [53] and Co–Cr–Fe–Mn–Ni high-entropy alloy [54], respectively. The dynamically recrystallized grain size increased with the growth of temperature, however, exhibited a non-linear relationship with strain rate for the ferritic stainless steel [14].

*4.5. Hot Flow Stress Model*

Based on the prediction of strain $e_p = f(Z)$, the material constants in the phenomenological constitutive model describing the flow stress $\sigma$ [MPa] depending on temperature $T$ [K], true strain rate $\dot{e}$ [s$^{-1}$] and true strain $e$ [-] were determined by multiple non-linear regression in Unistat 6.5 Statistics Software (Unistat Ltd., London, UK). This model was originally designed only for the smaller strains (before the steady-state region) but is also applicable on the relatively flat shape of stress–strain curves [12,67,77]. In the case of the high temperature range (i.e., at least 770 °C) and after the necessary reduction of the deformation range to a maximum value of 0.50, the equation was derived:

$$\sigma = 8751 \cdot e^{0.22} \cdot \exp\left(-0.22 \cdot \frac{e}{e_p}\right) \cdot \dot{e}^{(0.35 - \frac{296}{T})} \cdot \exp(-0.0030 \cdot T) \tag{9}$$

The accuracy and credibility of the derived model are evidenced by the relatively favourable values of the standard deviation (6.2 MPa) and the coefficient of determination ($R^2 = 0.9947$), as well as the graphical comparison in Figure 26. Here, on the horizontal axis, there is a true strain in the range 0.02–0.50 for each flow curve. From left to right, the individual curves correspond to the strain rates (all in s$^{-1}$) as follows: for a temperature of 1280 °C, $\dot{e}$ = 0.02–0.20–2.0–23; for a temperature of 1140 °C, $\dot{e}$ = 0.02–2.1–24; for a temperature of 1030 °C, $\dot{e}$ = 0.02–0.20–23; for a temperature of 930 °C, $\dot{e}$ = 0.02–0.20–22; for a temperature of 840 °C, $\dot{e}$ = 0.02–0.20–2.0–22; for a temperature of 770 °C, $\dot{e}$ = 0.02–2.0–23.

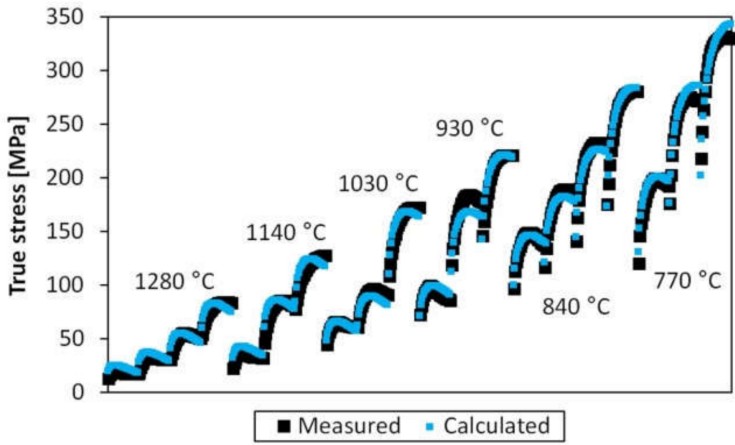

**Figure 26.** Accuracy of prediction of individual hot flow curves by Equation (9).

For the reliability of the developed model, a uniform distribution of deviations depending on individual deformation parameters is essential; the deviations are calculated as differences of measured and predicted stress values. The graphs in Figure 27a,c prove that Equation (9) meets this requirement very well in the case of deformation temperature and strain rate. Some stress–strain curves show systematically larger deviations in the whole range of strain (see Figure 27b) which is the result of usual data scattering when testing the hot deformation behaviour of metallic materials.

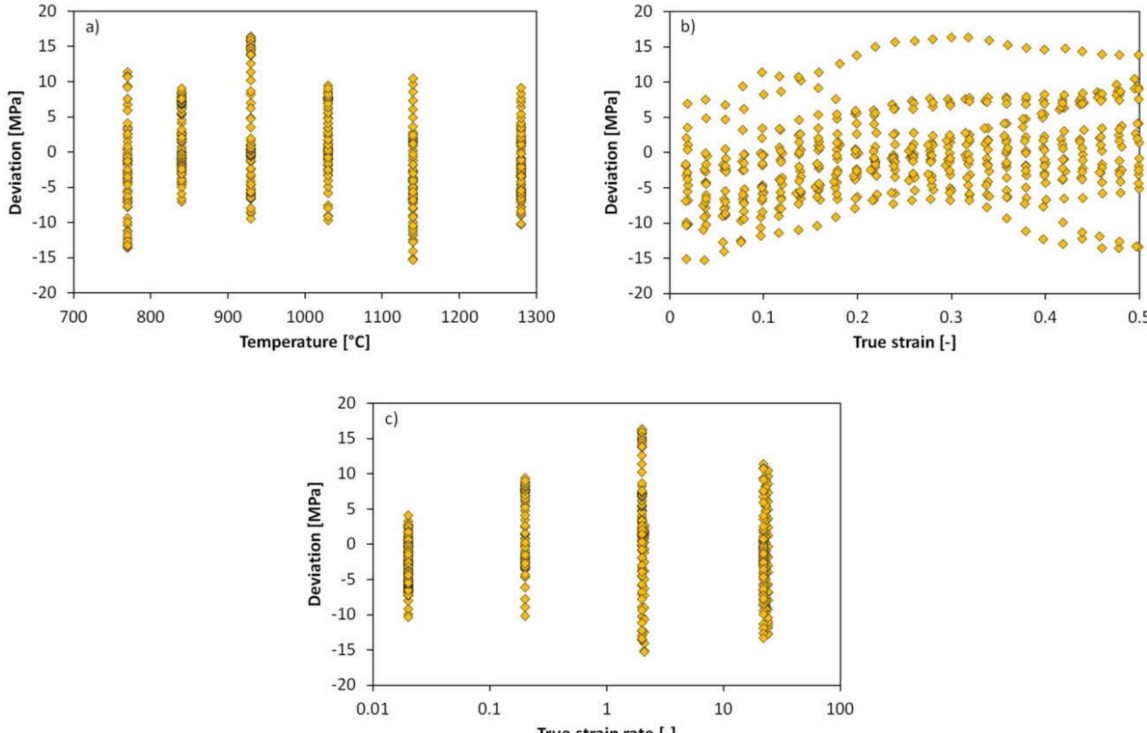

**Figure 27.** Distribution of flow stress deviations when applying Equation (9) to the experimental data depending on the parameters of individual tests. (**a**) Influence of temperature; (**b**) influence of strain; (**c**) influence of strain rate.

A similar accuracy was obtained when the general relationship corresponding to Equation (9) was applied to the AZ 31 magnesium alloy [67], the 6082 aluminium alloy [12] or grade C45 steel [78]. This physically based constitutive model has been shown to give similarly accurate results as the much more complex Hensel–Spittel phenomenological model [79]—see [78] for example.

Constitutive equation describing the flow behaviour of the bainite Mn–Si–Cr low-alloy steel with 0.58% C was developed for the temperature range of 950–1150 °C, i.e., for the austenite region [80]. A strain-compensated Arrhenius-type model was applied considering the fact that the material constants are in fact dependent on the strain value. The single-peak stress-strain curves have a similar shape to those in Figures 12 and 13. Deformation behaviour in the bainite region has not been investigated.

## 5. Conclusions

Up to a heating temperature of 757 °C, the microstructure of the as-cast medium-carbon steel alloyed with 1.2% Mn, 0.8% Cr and 0.2% Mo was formed mainly by bainite with ferritic and pearlitic networks occasionally lining the prior austenitic grain boundaries. At temperatures above 820 °C, the structure was completely austenitic.

For a strain rate ranging from 0.02 $s^{-1}$ to 20 $s^{-1}$ it was possible to describe peak stress as a function of the Zener–Hollomon parameter $Z$ with good accuracy. The calculated hot deformation activation energies $Q$ = 648 kJ·$mol^{-1}$ and $Q$ = 364 kJ·$mol^{-1}$ have been used for the temperatures of 650–770 °C and 770–1280 °C, respectively. The $Q$ value is, therefore, about 1.8 times lower in the austenite region than in the region characterized by the majority of bainite. This corresponds to the considerable increase of flow stress in the low temperature range. The deformation behaviour at the temperature of 770 °C corresponded to the high-temperature austenitic region because only a minimal amount of ferrite (about 1.3%) appeared at the boundaries of the austenitic grains.

If austenite predominates in the structure, the stress–strain curves have the shape typical for dynamic recrystallization, but with a relatively indistinct peak. Low-temperature stress-strain curves, corresponding mainly to the bainitic structure, are not so flat and show a peak at small strains practically independent on the deformation parameters; the stress drop behind the peak is very pronounced and associated with dynamic recrystallization of ferrite, which was demonstrated by a SEM analysis in the samples tested at a temperature of 650 °C. For plastic deformation in the austenitic region, the relationship between the peak strain and the parameter *Z* was derived, and with its use the phenomenological constitutive model describing the flow stress depending on temperature, true strain rate and true strain (up to *e* = 0.5 only) could be assembled. The model can be used to predict forming forces in the production of seamless tubes from the given steel.

After the strain *e* = 1.0, the size of the dynamically recrystallized austenitic grain strongly decreases with an increasing strain rate and decreasing temperature. The very low strain rate of 0.02 s$^{-1}$ usually does not lead to refinement of the prior austenitic grain, when the dynamically recrystallized grain can be significantly coarsened during the long time of deformation.

**Author Contributions:** I.S. coordinated the experimental activities, processed the results and wrote the first draft of the manuscript; P.O. performed the experiments and developed the constitutive model; P.K. performed the experiments and calculated the activation energy values; J.S. evaluated the results of structural analysis; K.K. performed the SEM-EDS analyses; S.R. contributed to the processing and discussion of results; R.K. performed the experiments; M.S. performed a literary analysis and editing; P.T. provided the material and its characterization; all authors revised and approved the final version of the manuscript. All authors have read and agreed to the published version of the manuscript.

**Funding:** The article was created thanks to the project No. CZ.02.1.01/0.0/0.0/17_049/0008399 from the EU and CR financial funds provided by the Operational Programme Research, Development and Education, Call 02_17_049 Long-Term Intersectoral Cooperation for ITI, Managing Authority: Czech Republic—Ministry of Education, Youth and Sports; and as part of internal grant projects SP2020/88 and SP2020/39 supported at VŠB—TU Ostrava by the Ministry of Education, Youth and Sports of the Czech Republic.

**Acknowledgments:** The authors would like to thank Ivana Malá (VŠB—TU Ostrava) for performing the light optical microscopy and metallographic analyses.

**Conflicts of Interest:** The authors declare no conflict of interest.

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
