# Peer review of "Hot Deformation Behaviour of Mn–Cr–Mo Low-Alloy Steel in Various Phase Regions"

_metals, doi:10.3390/met10091255_

Round 1
Reviewer 1 Report
Please look again the captions of figures.
- Figure 2: "Region of the prior austenite grain boundary with ferrite(F), pearlite(P) and bainite(B),----"
- Figure 8: What is "M"? What is "S"?
- Figure 16: What is open triangle? What is solid triangle? What is solid rhombus?
- Figure 22: "(a) flattened bainite aggregates(b) and recrystallized ferrite grains(F),---"
Reviewer 2 Report
This manuscript study the hot deformation behaviors of a Mn-Cr-Mo alloy with different temperature and strain rate.
1 The langue of this manuscript is poor. It is hard to read. For example: Page 1 line 22: For forming in the austenitic region, the relationship between the peak strain and Zener........
2 The manuscript is more like a lab report than a research paper. The authors should compare your results with others and an in-depth discussion on the experimental results should be given.
Reviewer 3 Report
This paper studies the flow stress and hot deformation activation energy of the medium-carbon steel of low alloyed in a wide range of conditions as well as to compare the deformation behaviour in the high-temperature region of austenite and in the low-temperature region associated with the occurrence of other structural components, especially bainite. Although the work seems interesting it should be modified before publishing it.
The authors of this paper must review English – the paper has some grammatical mistakes.
Abstract doesn’t define the aim clearly and novelty of paper also needs to be clarified on the abstract. The same should be added in the introduction.
Introduction
Line 43 - “Bainite steels are a permanent object of interest for researchers, but in a different sense” – This sentence should be rewritten, it is difficult to understand it.
Experimental Material and Procedures
The characterization of the steel carried out in this section is already a result; this should be explained in the text, since it is a bit confusing when following the document.
Line 84 -Where was the steel gotten? The authors should specify supplier, city and country.
Figure 3 should be explained more clearly.
Results
Line 140 and 143– What is Ac1? and Ac3? Although it is obvious, it must be defined in the text
Figure 8, the three figures must be named and explained in the text.
Section 3.2. How did the experimental true stress vs true strain curves get? It is not clear in the section experimental procedure.
Every figures are not sufficiently explained, there are many figures and I find it difficult to understand the process. The authors neither explain it clearly from the beginning. It should be rewritten.
Why are standard deviations too high in table 2?
Round 2
Reviewer 2 Report
I think the authors have done the revisions according to the comments and can be accepted.
Reviewer 3 Report
The paper can be published now in current format, but before the authors must review the line 130, there is a mistake.
Figures 12, 13 and 14 a new strain rates are included at 19, 22 and 23 ºC s-1 instead of 20. I supposse this is not a mistake, it is a little mismatch in the furnace, is not it?